# Prevalence of pectus excavatum in an adult population-based cohort estimated from radiographic indices of chest wall shape

**Mikaela Biavati[1], Julia Kozlitina[1], Adam C. Alder[2], Robert Foglia[2], Roderick W. McColl[3], Ronald M. Peshock[3], Robert E. Kelly, Jr[4], Christine Kim Garcia[1,5]***

**1** Eugene McDermott Center for Human Growth and Development, University of Texas Southwestern Medical Center, Dallas, TX, United States of America, **2** Department of Surgery, University of Texas Southwestern Medical Center, Dallas, TX, United States of America, **3** Department of Radiology, University of Texas Southwestern Medical Center, Dallas, TX, United States of America, **4** Department of Surgery and Pediatrics, Eastern Virginia Medical Center and Children's Hospital of the King's Daughters, Norfolk, VA, United States of America, **5** Department of Internal Medicine, University of Texas Southwestern Medical Center, Dallas, TX, United States of America

* ckg2116@cumc.columbia.edu

**Data Availability Statement:** All relevant data are within the paper and its Supporting Information files.

## Abstract

### Background

Pectus excavatum is the most common chest wall skeletal deformity. Although commonly evaluated in adolescence, its prevalence in adults is unknown.

### Methods and findings

Radiographic indices of chest wall shape were analyzed for participants of the first (n = 2687) and second (n = 1780) phases of the population-based Dallas Heart Study and compared to clinical cases of pectus (n = 297). Thoracic computed tomography imaging studies were examined to calculate the Haller index, a measure of thoracic axial shape, and the Correction index, which quantitates the posterior displacement of the sternum relative to the ribs. At the level of the superior xiphoid, 0.5%, 5% and 0.4% of adult Dallas Heart Study subjects have evidence of pectus excavatum using thresholds of Haller index >3.25, Correction index >10%, or both, respectively. Radiographic measures of pectus are more common in females than males and there is a greater prevalence of pectus in women than men. In the general population, the Haller and Correction indices are associated with height and weight, independent of age, gender, and ethnicity. Repeat imaging of a subset of subjects (n = 992) demonstrated decreases in the mean Haller and Correction indices over seven years, suggesting change to a more circular axial thorax, with less sternal depression, over time.

### Conclusions

To our knowledge, this is the first study estimating the prevalence of pectus in an unselected adult population. Despite the higher reported prevalence of pectus cases in adolescent boys, this study demonstrates a higher prevalence of radiographic indices of pectus in adult females.

**Funding:** The study was supported by grants UL1TR000451 and KL2TR000453 from the National Center for Advancing Translational Sciences as well as institutional funds. The funders had no role in study design, data collection and analysis, decision to publish, or preparation of the manuscript.

**Competing interests:** The authors have declared that no competing interests exist.

## Introduction

Pectus excavatum, or "funnel chest," results from the posterior displacement of the sternum and adjoining ribs into the thoracic cavity. It is the most common anterior chest wall deformity, which has been estimated to have an occurrence of 1 in 40 to 1 in 400 individuals across different cohorts[1–5]. Males are referred for evaluation 3–5 times more often than females[6]. Pectus carinatum, or "pigeon chest," is the second most common chest wall deformity and is characterized by the anterior protrusion of the sternum and the adjoining ribs. Although pectus can be present at birth, most patients experience a marked worsening in the severity of their sternal deformity in adolescence during their pubertal growth spurt[7]. When severe, pectus excavatum leads to dyspnea, pulmonary restriction, compression of the heart and great vessels and psychological distress. Cardiopulmonary limitations are generally considered to be related to the severity of the pectus deformity[8]. Severe cases of pectus excavatum can be corrected surgically, with either the open or Ravitch procedure or the minimally invasive Nuss procedure[9, 10]. Pectus carinatum can be corrected by surgery or the application of a compressive chest brace[11].

Radiographic measures can quantify thoracic skeletal abnormalities and influence decisions regarding the need for surgical repair. One standard metric is the Haller index, which is the quotient of the internal transverse thoracic distance and the internal anteroposterior dimension at the level of the deepest point of the sternal deformity[12]. A Haller index greater than 3.25 is considered an indication for surgical evaluation. Another quantitative measure is the Correction index, which is a measure of the posterior depression of the sternum relative to the adjacent ribs[13]. A Correction index of greater than 10% is considered to be indicative of substantial pectus excavatum[13].

Although the prevalence of pectus in birth and pediatric cohorts has been studied[1, 14, 15], its prevalence in adult population-based cohorts is unknown. The purpose of this study was to quantify radiographic measures of thoracic shape in the Dallas Heart Study (DHS), a large adult, multiethnic, population-based cohort, to estimate the prevalence of pectus excavatum in adults. To decrease bias, we calculated measures of thoracic shape at three skeletal landmarks (T6, T8, and Superior Xiphoid). Thoracic imaging of pectus cases were used for comparison.

## Methods

### Study populations

This study was approved by the University of Texas Southwestern Medical Center and the Eastern Virginia Medical Center Institutional Review Boards. Written informed consent was obtained from all subjects. All studies were performed in accordance with relevant guidelines and regulations. The DHS is a longitudinal, multiethnic, population-based probability sample of Dallas County residents. Details of the study design have been described previously[16]. The study was initiated in 2000 and transformed from a cross-sectional study to a longitudinal study in 2007. Some participants from the first study phase (DHS1) underwent repeat evaluation. Participation of unrelated spouses and friends of the original cohort augmented the follow-up DHS2 cohort (NCT00344903). A subset of the original DHS1 cohort (n = 2687) had cardiac electron beam computed tomographic scans (Imatron EBCT Scanner) available for analysis. A subset of DHS2 participants (n = 788 unrelated and n = 922 prior DHS1 participants) had cardiac multi-detector computerized tomographic scans (Toshiba Aquilon 64-slice MDCT) available for analysis.

The cases referred for evaluation of pectus were obtained by Children's Hospital of The King's Daughters/Eastern Virginia Medical School from 2009–2017 and by Children's Medical

Center of Dallas/University of Texas Southwestern Medical Center from 2014–2017. Cases included those diagnosed with a chest wall defect by a pediatric surgeon. Only those cases with an available thoracic computed tomography (CT) scan were included in this study, representing 58% cases from Children's Hospital of the King's Daughters/Eastern Virginia Medical School and 17% cases from Children's Medical Center of Dallas/University of Texas Southwestern Medical Center. The cases included those with pectus excavatum, carinatum or a complex mix of both (mixed deformity). Patients were treated with a variety of surgical or non-surgical (brace, vacuum bell, or other) approaches.

### Quantitation of sternal deformity

Scans were excluded if there was evidence of chest trauma or prior thoracic surgical procedure or if skeletal landmarks were not visible. Axial (transverse) images of the thorax were inspected. The indices were calculated at the levels of the mid-vertebral T6 and T8 as well as the superior xiphoid. An additional measurement was calculated at the point of maximal sternal depression or maximal sternal protrusion.

The Haller index[12] was calculated as the quotient of the greatest transverse dimension of the chest (A) and the minimal anterior-posterior dimension from the vertebral body or its horizontal tangent to the posterior sternum (B), that is, Haller index = A/B (**Fig 1B**). In cases of asymmetric pectus excavatum the anterior-posterior dimension was measured from the vertebral body (or its horizontal tangent) to the most posterior depression of the costal cartilage adjacent to the sternum. The Correction index[13] measurements include the anterior-posterior distance between the vertebral body and the sternum or the most posterior depression of the adjacent costal cartilage (B) as well as the maximal distance from the horizontal tangent of the vertebral body to the inner margin of anterior chest cavity (C). It was calculated as Correction index = [(C–B)/C] as shown in **Fig 1C**. For pectus carinatum cases, we drew a line posterior to the sternum connecting the right and left hemithoraces that followed the curvature of the thorax; line segment (C) was measured between the intersection of this line with the anterior chest and the horizontal tangent of the vertebral body. All distance measurements were made using the ImageJ ruler tool.

### Statistical methods

Data are summarized as number and percentage or median and interquartile range (IQR). Measures of sternal deformity were compared between demographic groups using Wilcoxon rank-sum test. The prevalence of pectus was compared between groups using Fisher's exact test. The relationship of Haller and Correction indices with age (years), height (cm), weight (kg), and BMI (kg/m$^2$) was first examined using simple (unadjusted) linear regression. Multivariable-adjusted linear regression models were used to examine the association of chest wall shape with demographic and anthropometric characteristics. Repeated measures of Haller and Correction indices in the DHS participants were compared using Wilcoxon signed-rank test. We applied an inverse normal transformation to Haller and Correction indices prior to fitting linear models to achieve approximate normality of the residuals.

### Results

#### Assessment of chest wall abnormalities of 297 pectus cases

The Haller and Correction indices were calculated from axial images of thoracic CT scans of patients referred for surgical evaluation of pectus (n = 297). The cases were mostly men (78%), Non-Hispanic White (91%) and adolescent, with a median age of 15 years (**S1 Table**). We

**A.**

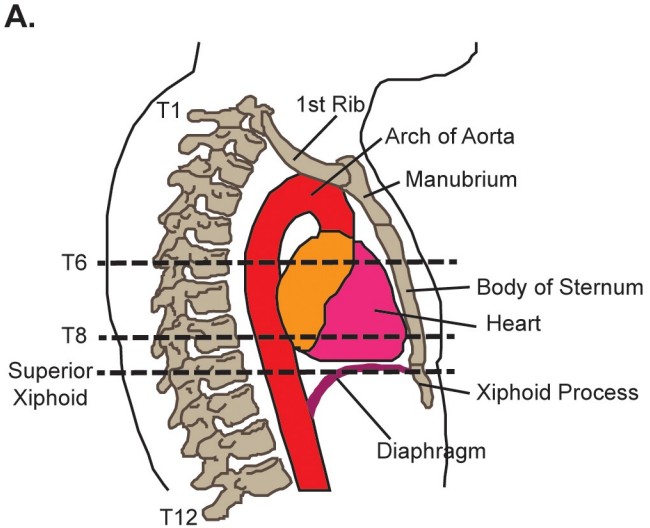

**D. Haller Index Measurements**

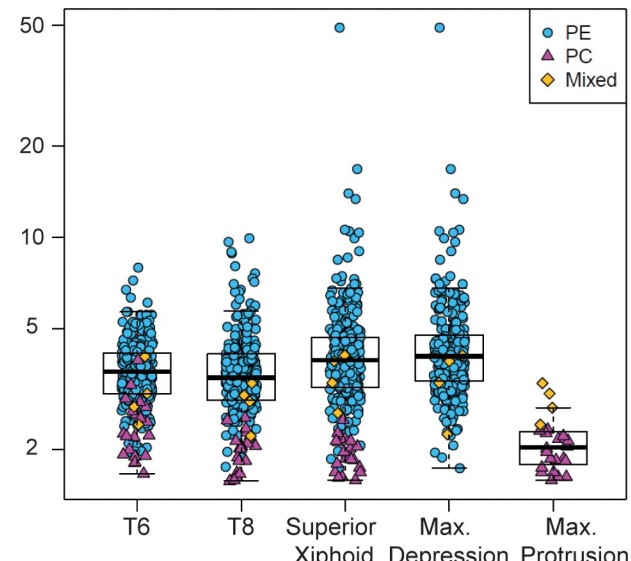

**B. Haller Index Measurements**

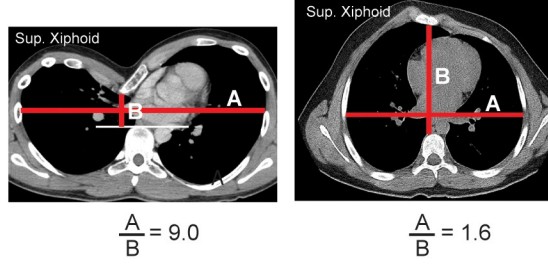

**C. Correction Index Measurements**

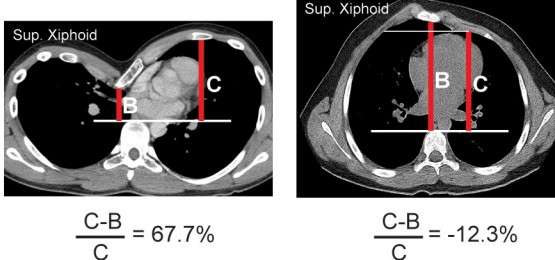

**E. Correction Index Measurements**

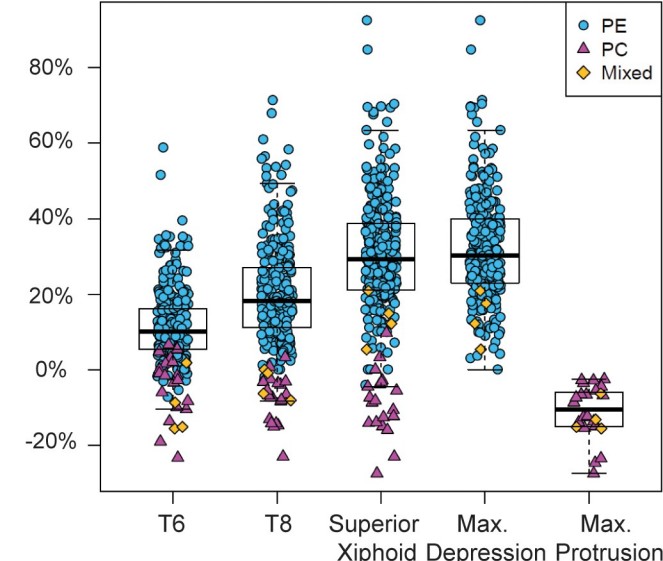

**Fig 1. Calculation of the haller index and correction index from axial images of chest computed tomography scans of pectus cases.** Anatomic schematic of the lateral view of the thorax (**A**) demonstrating the position of the T6 and T8 vertebral bodies relative to the body of the sternum and the superior xiphoid process. Measurements of chest wall dimensions (red lines) used to calculate the Haller Index (**B**) and Correction Index (**C**) from a patient with pectus excavatum (left) and a patient with pectus carinatum (right). Measurements in B and C are shown at the level of the superior xiphoid. Haller (**D**) and Correction (**E**) Index measurements of patients with pectus excavatum (PE, n = 274, blue circles), pectus carinatum (PC, n = 19, red triangles) and those with a mixed pectus excavatum and carinatum defect (Mixed, n = 4, orange diamonds) at the level of T6, T8, Superior Xiphoid, as well as the point of maximal (max.) sternal depression (n = 278) or protrusion (n = 23). Patients with a mixed defect are included in both the maximal sternal depression and protrusion data sets. Box and whisker plots (black) are superimposed.

calculated indices for cases (pectus excavatum, n = 274; pectus carinatum, n = 19; mixed deformity, n = 4) at the level of T6, T8, and the superior xiphoid as well as the point of maximal sternal depression or protrusion (**Fig 1**). The Haller index measurements at T6, T8 and the superior xiphoid do not differ much from that measured at the point of maximal depression

(Fig 1D). In contrast, the Correction index at the site of maximal deformity correlates best with that measured at the level of the superior xiphoid (Fig 1E), consistent with the observation that the deepest part of the pectus depression usually affects the caudal portion of the sternum. The female pectus cases had a greater degree of severity of excavatum than the male cases, as determined by Haller and Correction indices at four different axial levels (T6, T8, Superior Xiphoid and the point of maximal depression) (Fig 2, S2 Table). A Correction index of >10% was found at the point of maximal depression in 96% of the PE cases, whereas, a Haller index of >3.25 was found in fewer (78%) cases (Fig 3A).

## Measurement of chest wall shape in a multi-ethnic population-based cohort

Thoracic imaging was analyzed for the Dallas Heart Study, a multi-ethnic, adult population based cohort from Dallas, Texas. The subjects included those who participated in the initial study (DHS1) (n = 2687) as well as the follow up study (DHS2) 6–7 years later. Participants in DHS2 included a subset from DHS1 who had repeat imaging (n = 992) and an unrelated group who were newly enrolled (n = 788) (S1 Table).

The Haller index measures the ratio of the longitudinal to transverse dimension of the thorax in the axial plane. A smaller Haller index correlates to a more circular shape and a larger one correlates with a more oval shape. A larger Haller index, thus, a more oval shape, is found in females as compared to males at the T6, T8 and superior xiphoid in DHS1 (Fig 2A, S2 Table). The Correction index measures the position of the sternum relative to the lateral rib cage. A larger Correction index indicates a more "sunken" position of the sternum relative to the lateral ribs. A negative Correction index indicates a more "peaked" position of the sternum. A larger Correction index, thus, a more "sunken" position of the sternum is seen in females as compared to males at all axial levels in DHS1 (Fig 2B, S2 Table). Although the magnitude of change is small in the DHS, these findings are consistent at multiple levels (T6, T8 and Superior Xiphoid) and in both the DHS1 and the DHS2 cohorts (S2 Table). In general, females have a more oval axial thoracic shape with a more sunken position of the sternum.

## Prevalence of severe pectus excavatum in population-based cohort

There is more separation between the cases and population-based controls at the level of the superior xiphoid than at T6 or T8 (Fig 2). When the Haller index is plotted against the Correction index, the cases with pectus excavatum are distinguished from those with carinatum at the level of maximal sternal deformity (Fig 3A) and the superior xiphoid (Fig 3B). Most pectus excavatum cases have a Haller index >3.25 and a Correction index >10%. When we apply these same thresholds to the DHS1 cohort, we find that 0.5% have a Haller index >3.25 and 5% have a Correction index >10% (Fig 3C, Table 1). When we use a combination of these thresholds to define pectus excavatum at the level of the superior xiphoid, we find a prevalence of 0.4%, or 1 in 250 individuals, in the DHS1 cohort. A higher prevalence is found in women (0.5%; 1 in 200) than men (0.3%; 1 in 333). Similar estimates of prevalence are found in the DHS2 cohort that incudes unrelated subjects (Table 1) or when assessed at T6 or T8 (S3 Table).

## Chest wall measurements independently associated with ethnicity, age and weight

The chest wall shape of the Dallas Heart Study cohorts differ by ethnicicty (S4 Table). In general, Non-Hispanic blacks have a smaller CI than non-Hispanic whites or Hispanics,

## A. Haller Index Measurements

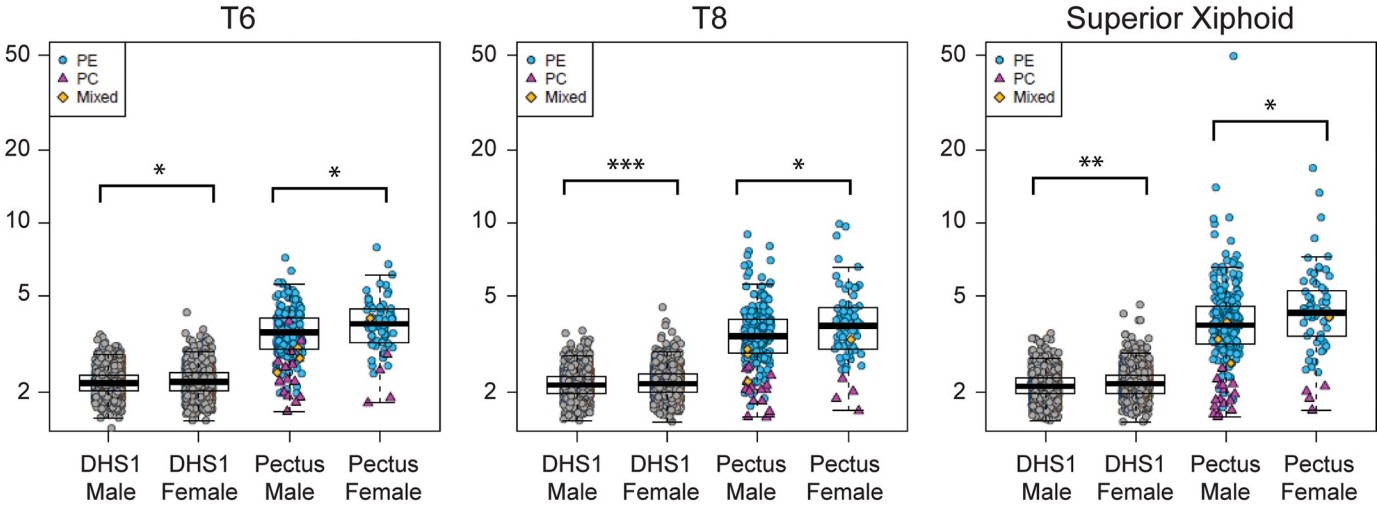

## B. Correction Index Measurements

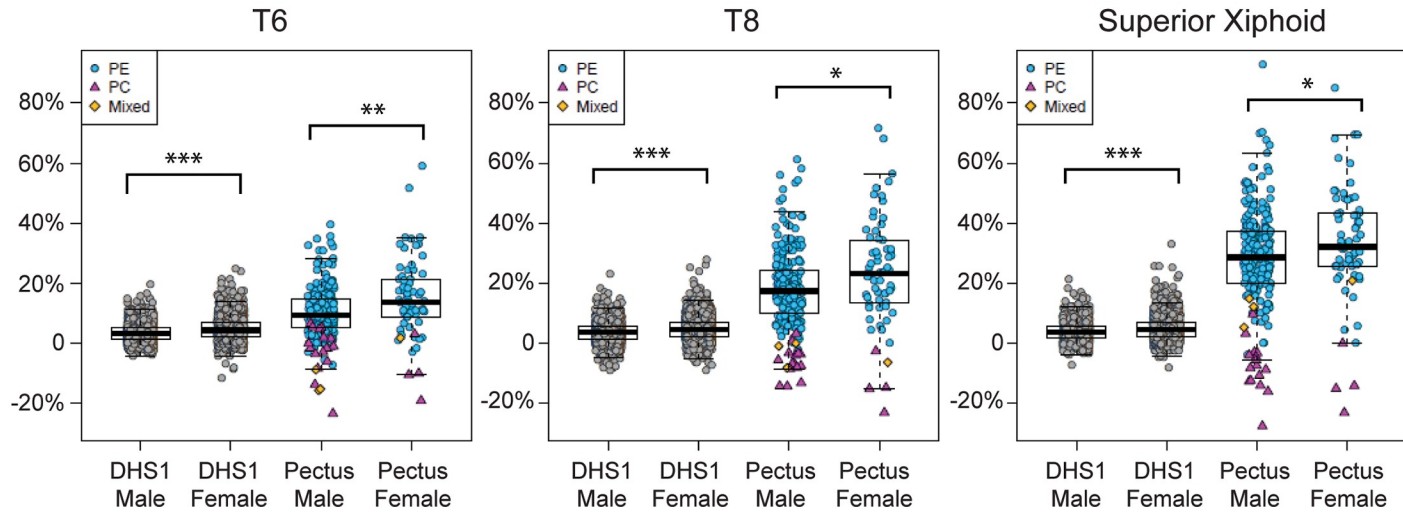

**Fig 2. Comparison of haller and correction index measurements for men and women from the Dallas Heart Study (DHS) and pectus cohorts.** Haller (**A**) and Correction (**B**) indices of the DHS1 cohort (n = 2687) and pectus cases (n = 297) at T6, T8 and superior xiphoid axial levels. Individuals from DHS1 (gray circles) and subjects with pectus excavatum (PE, n = 274, blue circles) pectus carinatum (PC, n = 19, red triangles) or a mixed pectus excavatum and carinatum defect (Mixed, n = 4, orange diamonds) are individually plotted with superimposed box and whisker plots (black). *, ** and *** indicative of P-value <0.05, <0.001 and <0.0001, respectively.

indicating less likelihood of a posteriorly-depressed sternum. These findings were seen in both the DHS1 and DHS2 cohorts. Although not consistent across all axial levels, Non-Hispanic Blacks tend to have a lower Haller index than whites in the DHS1 cohort. This suggests a larger anterior-posterior depth relative to the lateral dimension, or a more circular rather than oval axial thoracic shape.

We find that there are significant associations between chest wall shape, age and weight in the DHS1 cohort (**Fig 4A and 4B**). There is a negative association between the Haller index and weight (p<0.0001), BMI (p<0.0001) and age (p<0.0001). There is a similar negative association between the Correction index and weight (p<0.0001) or BMI (p<0.0001). In contrast

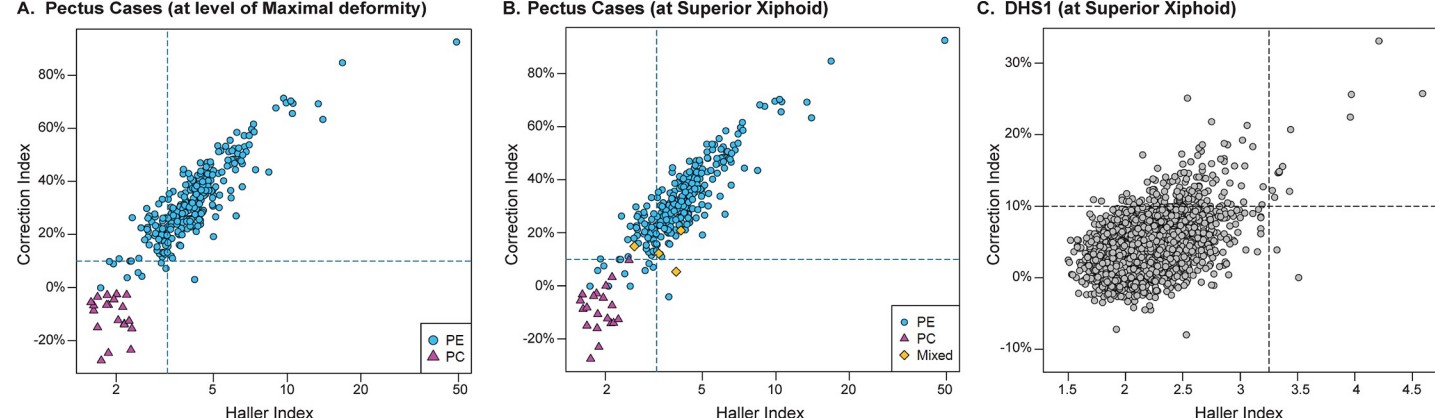

**Fig 3. Correlation of the correction index with the haller index for pectus and Dallas Heart Study (DHS) cohorts.** The Correction and Haller index at the level of the superior xiphoid is plotted for each of the pectus excavatum (PE, n = 274, blue circles), pectus carinatum (PC, n = 19, red triangles) and mixed pectus excavatum and carinatum (Mixed, n = 4, orange diamonds) patients (**A, B**) and for each of the DHS1 subjects (**C**) (n = 2687). Indices are measured at the point of maximal sternal deformity, that is, at the point of maximal sternal depression for the PE cases and the point of maximal sternal protrusion for the PC cases (**A**), or at the superior xiphoid (**B, C**). Dashed lines indicate a Haller Index of 3.25 or a Correction Index of 10%. Note the different axes.

with the Haller index, the Correction index was positively, but modestly, correlated with age (p = 0.0045).

Multivariable models of chest wall shape of DHS1 demonstrate independent associations of ethnicity, gender, age, height and weight with the Haller and Correction indices across different thoracic levels (**S5 Table**). The indices are associated with height (β [SE], 0.029(0.003), p<0.0001 and 0.022(0.003), p<0.0001, respectively) and weight (-0.024(0.001), p<0.0001 and -0.013(0.001), p<0.0001, respectively) at the superior xiphoid process, independent of age, gender, and ethnicity. Similar results are found when analyzing the unrelated DHS2 cohort (**S6 Table**).

**Table 1. Prevalence of pectus excavatum in cases and population-based cohorts at the level of the superior xiphoid.**

| Population | Pectus Definition | Pectus Cases N | Pectus Cases N (%) | Male N | Male N (%) | Female N | Female N (%) |
|---|---|---|---|---|---|---|---|
| All Pectus Cases | HI>3.25 | 297 | 215 (72) | 231 | 164 (71) | 66 | 51 (77) |
| (including PE and PC) | CI>10% | 297 | 265 (89) | 231 | 206 (89) | 66 | 59 (89) |
| | HI>3.25 and CI>10% | 297 | 213 (72) | 231 | 162 (70) | 66 | 51 (77) |
| DHS1 | HI>3.25 | 2687 | 14 (0.5) | 1158 | 6 (0.5) | 1529 | 8 (0.5) |
| | CI>10% | 2687 | 143 (5) | 1158 | 43 (4) | 1529 | 100 (6) |
| | HI>3.25 and CI>10% | 2687 | 12 (0.4) | 1158 | 4 (0.3) | 1529 | 8 (0.5) |
| DHS2 | HI>3.25 | 788 | 4 (0.5) | 249 | 0 (0) | 539 | 4 (1) |
| (Not in DHS1) | CI>10% | 788 | 53 (7) | 249 | 7 (3) | 539 | 46 (8) |
| | HI>3.25 and CI>10% | 788 | 3 (0.4) | 249 | 0 (0) | 539 | 3 (0.6) |
| DHS2 | HI>3.25 | 992 | 7 (0.7) | 278 | 1 (0.4) | 714 | 6 (0.8) |
| (Subset of DHS1) | CI>10% | 992 | 77 (7.8) | 278 | 11 (4) | 714 | 66 (9.2) |
| | HI>3.25 and CI>10% | 992 | 7 (0.7) | 278 | 1 (0.4) | 714 | 6 (0.8) |

P-values calculated using Fisher exact test.

Abbreviations: Pectus Excavatum (PE), Pectus Carinatum (PC), Haller Index (HI), Correction Index (CI).

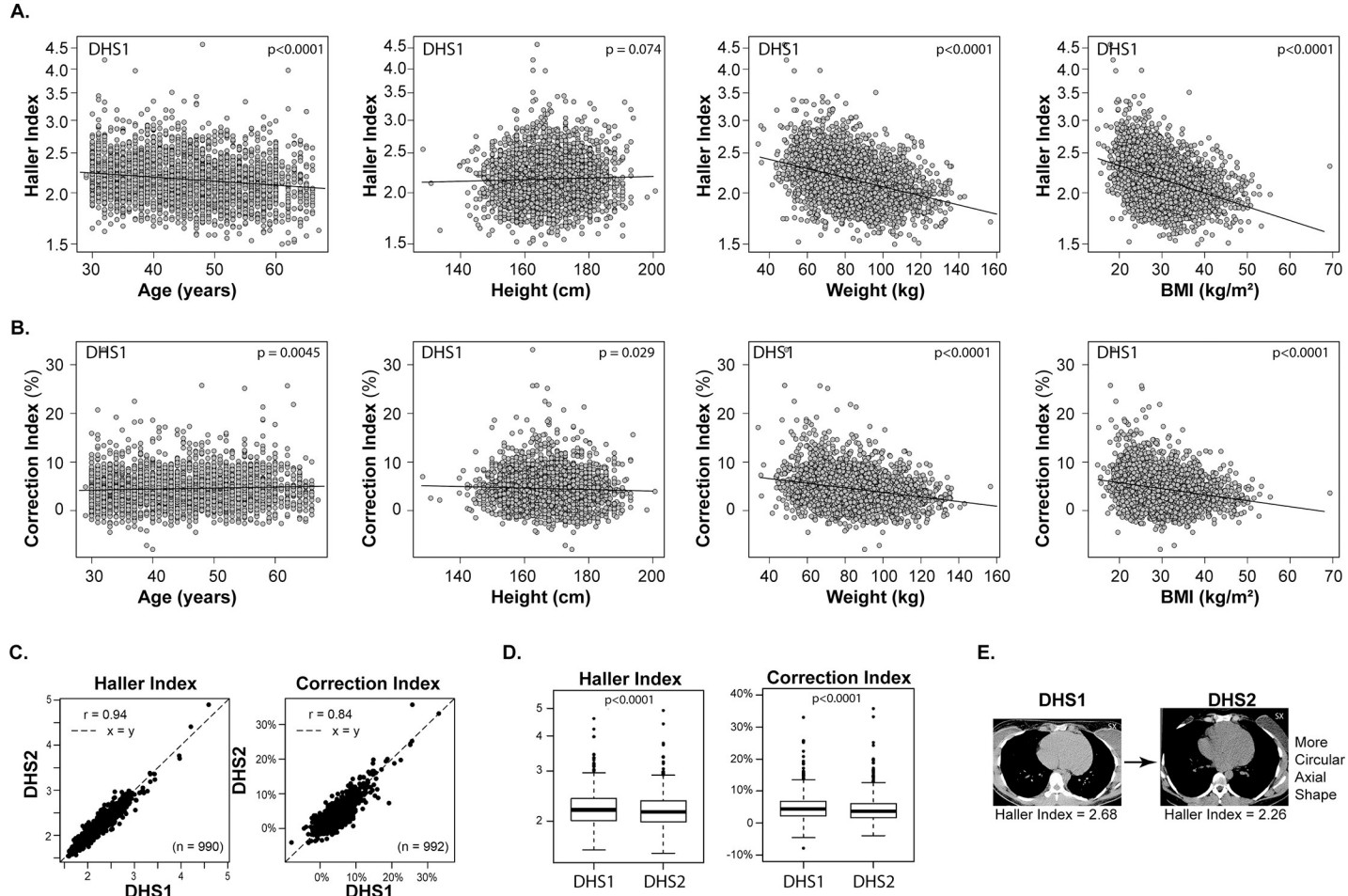

**Fig 4. Haller and correction indices of the Dallas Heart Study subjects.** Haller index (**A**) and Correction Index (**B**) measured at the superior xiphoid axial level is plotted against the age in years, height in cm, weight in kg and BMI in kg/m² for DHS1 (n = 2687) subjects. Individual subjects are represented as gray circles; the trend line is superimposed. Correlation (**C**) and box and whisker plots (**D**) of Haller and Correction indices for DHS Subjects (n = 992) with available imaging from DHS1 and DHS2. A DHS subject (**E**) demonstrating an increase in the Haller index in the repeat imaging study at the level of the superior xiphoid. The magnitude of Haller index increase is within the top 1%. Imaging shows the development of a more circular, and less oval, axial chest wall shape.

### Change in chest wall measurements in the population over 7 years

We analyzed chest wall measurements for the 992 subjects who underwent repeat imaging. The repeated measures were highly correlated (**Fig 4C**). There was a small but significant mean decrease in the Haller index from DHS1 to DHS2 (2.19 [2.01–2.40] vs 2.15 [1.99–2.35], p<0.0001) (**Fig 4D**), indicating a trend toward developing a more circular transverse chest wall shape. There was a decrease of the mean Correction index from DHS1 to DHS2 (4.5 [2.2–6.7] vs 3.6 [1.7–6.1], p<0.0001) indicating a trend toward a less depressed position of the sternum in repeat imaging.

## Discussion

Although pectus excavatum has been characterized as one of the most common birth defects [1, 17], its prevalence in adult populations is unknown. In this study we estimate the prevalence of pectus excavatum at ~0.4%, or 1 in 250 individuals, in a large, population-based, multiethnic, adult population with a median age of 44 years. We have used a combination of two,

independent and validated radiographic measures of chest wall shape, the Haller and Correction indices, to characterize cohorts of pectus clinical cases and unselected population-based controls. We find significant associations between these indices with age, gender, ethnicity, height and weight. It is striking that the radiographic measurements associated with pectus excavatum differ significantly by gender. Both the Haller and Correction indices are larger in females than males for both the case and control cohorts. Although the absolute numbers are small, we find that more females (0.5%) than males (0.3%) have pectus excavatum in the population cohort as defined by these indices. This finding is surprising since the literature suggests that a greater number of boys, usually 4-fold higher than girls, are referred for evaluation of pectus[6, 18]. Even in the current study, there are 3.5 times more males than females in the pectus case cohort. Thus, there is a strong referral bias to evaluate adolescent boys for pectus as breast development and modesty may mask the underlying chest wall defect in affected females. Future studies will need to determine the generalizability of these findings in unselected cohorts of different ages.

These observations would not be possible without the availability of chest CT scans that allow for unbiased calculation of quantitative indices. Several radiographic metrics have been studied to quantitate the severity of the pectus deformity. The Haller index is the most widely used, although it does not directly measure sternal depression, but rather, the quotient of the transverse and anterior-posterior axial dimensions. The original description of this index in 1987 was without reference to bony thoracic landmarks, but was calculated in the axial plane that generated the largest value[12]. Cross-sectional pediatric cohort studies have found that the Haller index tends to increase with age for the pediatric population, especially at the most cranial (T3-T5) and caudal (T11-T12) levels[14, 15]. Hence, prior studies suggest that the axial thorax tends to flatten from a more circular to a more oval shape during childhood. Perhaps this explains why prevalence of pectus excavatum is lower in a birth cohort (1:400)[1] than in school-aged children[2, 4, 5] (up to 1:40[3]). Also, perhaps this is why most individuals are referred for surgical evaluation of chest wall deformities during adolescence. Here, we find the opposite trend for Haller index with age; namely, it tends to decrease with advancing age in adults. Thus, this study suggests that the axial thorax tends to return to a more circular shape during adulthood.

Although the diagnosis of pectus does not require a chest CT scan, imaging studies provide quantitative measures of severity, thus allowing for comparison across studies. The original descriptions of the Haller and Correction indices suggested cut-offs of >3.25[12] and >10% [13], respectively, as thresholds to define pectus excavatum in children. Ten-fold fewer DHS1 subjects have a Haller index >3.25 than a Correction index >10% at the level of the superior xiphoid. By using a combination of both thresholds, we conservatively estimate the population-based prevalence of pectus excavatum in adults of 0.4%. In comparison, 78% and 96% of the pectus cases have a HI >3.25 or CI >10%, respectively, at the point of maximal sternal deformity. Given the difference in age between the cases and the population-based controls, we are not suggesting that the latter should provide guidance regarding cut-offs for evaluation or treatment of adolescents with pectus. However, the adult population does highlight the extreme nature of the cases evaluated by the surgical centers.

Although there is correlation between the Haller and Correction indices in pectus cases [19], the correlation between the two indices in population-based cohorts is unknown. We find that there is evidence for a more oval axial thorax shape and a greater degree of sternal depression in adult women than men. At three different axial levels, the indices are larger for Whites or Hispanics than Blacks. In light of this observation, it is interesting to note that Whites are the most highly represented amongst the pectus cases. At every axial level, a more oval axial thorax and a more depressed sternum is associated with taller height and decreased weight.

There are a number of limitations of the current study. Because of the cardiac-centric imaging of the DHS, the shape of the chest was measured solely in the mid chest. Only two indices

were measured, although others have been used in other studies to quantitate the pectus deformity[20]. Some of the variability in thorax shape in subjects from DHS1 to DHS2 may be due to differences in breath hold techniques[21, 22], although variability due to respiration tends to be less in the mid-chest[15]. Beside respiration, variability may be related to difficulty in identifying bony landmarks and human error. We have not used the axial image that represents the maximal deformity in the population-based cohorts, as is commonly done for evaluation of pectus cases, since this would impart less objectivity in evaluating mild abnormalities. Additional studies will be needed to correlate these quantitative indices with different dysmorphologic varieties of pectus, such as cup, saucer and trench shaped deformities[23].

Although association does not prove causation, the very strong inverse association between the Haller index and weight or BMI suggests that obesity may lead to a more circular axial thoracic shape. These observations are observed across different axial levels, for two independent indices of thoracic shape and in both unrelated subsets of the DHS cohort. Weight and BMI have also been found to be inversely related to severity of the pectus defect in an independent cohort[24]. Obese subjects are at greater risk for inspiratory muscle (i.e., diaphragm) fatigue at rest and with exercise[25–27]. Obesity also changes respiratory compliance and lung volumes. All of these factors may affect the shape of the thorax.

The clinical utility of the current study is that it provides a framework for comparing quantitative chest wall indices in pectus cases to an unselected, large, population-based, multiethnic population. The interval of time, 6 to 7 years, between the imaging studies evaluated in this study is small with regard to the subjects' lifespans. Additional studies may clarify the progression of thoracic changes over time from birth to adulthood for those with or without pectus. Identification of the factors that influence the temporal plasticity of the thorax may reveal more about the ontogeny of pectus and chest wall defects.

## Supporting information

**S1 Table. Demographic characteristics of participants.**
(DOCX)

**S2 Table. Haller and correction index measurements in pectus cases and DHS cohorts at multiple axial levels (T6, T8 and superior xiphoid) by gender.**
(DOCX)

**S3 Table. Prevalence of pectus excavatum in cases and population-based cohorts.**
(DOCX)

**S4 Table. Chest wall shapes in population-based cohorts by ethnicity.**
(DOCX)

**S5 Table. Linear models of the haller and correction index of the Dallas Heart Study (DHS1) cohort (n = 2685) at three axial levels (T6, T8 and superior xiphoid).**
(DOCX)

**S6 Table. Linear models of chest wall shape for the unrelated Dallas Heart Study (DHS2) cohort (n = 788) at three axial levels (T6, T8 and superior xiphoid).**
(DOCX)

## Acknowledgments

The authors thank Helen Hobbs, MD, and the other members of the Dallas Heart Study steering committee for access to the thoracic imaging studies.

## Author Contributions

**Conceptualization:** Mikaela Biavati, Christine Kim Garcia.

**Formal analysis:** Mikaela Biavati, Julia Kozlitina, Christine Kim Garcia.

**Investigation:** Mikaela Biavati, Julia Kozlitina, Adam C. Alder, Robert Foglia, Roderick W. McColl, Ronald M. Peshock, Robert E. Kelly, Jr.

**Methodology:** Mikaela Biavati, Julia Kozlitina, Robert E. Kelly, Jr, Christine Kim Garcia.

**Visualization:** Julia Kozlitina.

**Writing – original draft:** Christine Kim Garcia.

**Writing – review & editing:** Mikaela Biavati, Julia Kozlitina, Adam C. Alder, Robert Foglia, Roderick W. McColl, Ronald M. Peshock, Robert E. Kelly, Jr, Christine Kim Garcia.

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
