## [Decision Letter · Decision Letter 0]

28 Jan 2020

PONE-D-19-18160

Prevalence of Pectus Excavatum in an Adult Population-Based Cohort Estimated from Radiographic Indices of Chest Wall Shape

PLOS ONE

Dear Dr. Kim Garcia,

Thank you for submitting your manuscript to PLOS ONE. After careful consideration, we feel that it has merit but does not fully meet PLOS ONE’s publication criteria as it currently stands. Therefore, we invite you to submit a revised version of the manuscript that addresses the points raised during the review process.

We would appreciate receiving your revised manuscript by Mar 13 2020 11:59PM. To enhance the reproducibility of your results, we recommend that if applicable you deposit your laboratory protocols in protocols.io, where a protocol can be assigned its own identifier (DOI) such that it can be cited independently in the future. For instructions see: http://journals.plos.org/plosone/s/submission-guidelines#loc-laboratory-protocols

We look forward to receiving your revised manuscript.

Kind regards,

Agostino Chiaravalloti, MD, PhD

Academic Editor

PLOS ONE

Additional Editor Comments (if provided):

Dear Authors,

Reviewers have now commented on your paper. You will see that they are advising that you revise in a significant way your manuscript.

Journal Requirements:

Reviewers' comments:

Reviewer's Responses to Questions

**Comments to the Author**

1. Is the manuscript technically sound, and do the data support the conclusions?

Reviewer #1: Yes

Reviewer #2: Partly

Reviewer #3: Yes

2. Has the statistical analysis been performed appropriately and rigorously? 

Reviewer #1: I Don't Know

Reviewer #2: N/A

Reviewer #3: Yes

3. Have the authors made all data underlying the findings in their manuscript fully available?

Reviewer #1: Yes

Reviewer #2: Yes

Reviewer #3: Yes

4. Is the manuscript presented in an intelligible fashion and written in standard English?

Reviewer #1: Yes

Reviewer #2: Yes

Reviewer #3: Yes

5. Review Comments to the Author

Reviewer #1: This study provided an overview for the prevalence of pectus excavatum (PE) in the general adult population. The measurement protocol is simple, and the large sample size makes for a decent estimation. Currently, PE is most mentioned in neonates and children. Therefore, there lack existing evidence in the literature to effectively evaluate the accuracy of this study. On the other hand, however, this study has the potential to become one of the first in its kind, and could pave the way to future studies in adult PE. There are a few comments that I would like the authors to address.

1.As shown in Table 1, while evaluating the diagnostic indices of PE, was it necessary to include cases with pectus carinatum (PC)? The likelihood of decreased Haller index in PC patients might cause an underestimation in terms of the cases that fits the criteria.

2.What was the diagnostic criteria for the reference pectus cases? Apparently not all referred cases fit both the HI and CI criteria. Providing the detailed diagnostic criteria of these cases could increase the credibility of this sample.

3.While evaluating the paired samples in DHS1 and DHS2, with such a large sample size, parametric tests are usually preferred due to their higher power. Did the samples not follow normal distribution? If so, the possibility of the sample being biased may need to be addressed.

4.The estimated prevalence of undiagnosed, untreated adult pectus excavatum (1 in 250) was higher than the assumed incidence of neonatal pectus excavatum, 1 in 400, in 1975, where CT was much less available. Do the authors have any theory in regard to this phenomenon?

5.What is the clinical significance of this study? Do the authors have any expectation as to what changes this study might lead to in the management of adult PE?

Reviewer #2: The authors present a retrospective analysis of thoracic computed tomography imaging studies, reviewing the Haller index, a measure of thoracic axial shape, and the Correction index. They found that very strong inverse association between the Haller Index and weight or BMI suggests that obesity may lead to a more circular axial thoracic shape.

Overall, the most important problem of this manuscript is its novelty. Similar studies have been done by other investigators before, with the same modality or other modality both. Just to list a few:

Incidence and Classification of Chest Wall Deformities in Breast Augmentation Patients.

Aesthetic Plast Surg (United States), Dec 2017, 41(6) p1280-1290

Relationship between cardiac MR compression classification and CT chest wall indexes in patients with pectus excavatum. J Pediatr Surg (United States), Nov 2018, 53(11) p2294-2298

Szafer D, Taylor JS, Pei A, et al.

A Simplified Method for Three-Dimensional Optical Imaging and Measurement of Patients with Chest Wall Deformities. J Laparoendosc Adv Surg Tech A (United States), Feb 2019, 29(2) p267-271

Reviewer #3: INTRODUCTION

Page 3:

First reference [1] is from 1975! Is there no newer data available?

Page 3:

You describe that radiographic measures can quantify …; but is it always necessary for diagnosis? What added value does CT (radiation exposure) have, especially for patients in adolescence. A small paragraph like diagnosis is based on … would be helpful.

Page 3:

A Correction Index of 10% - do you mean exactly or greater than?

METHODS

Page 4:

The cases referred for evaluation of pectus …; During which period were the patients selected? How many cases were obtained and how many were included in the study? Why did all patients undergo a CT?

Page 5:

Definition of Haller and Correction Index thresholds - taken from the literature?

RESULTS

Page 7:

The Haller Index measures …

The Correction Index measures …; just a repetition of the introduction / methods - can be shortened.

Page 8:

You describe significant associations; do the parameters mentioned have a predictive value?

Page 9:

correlated with age (p=0.0045); for real? Or 0.045?; if you look at the corresponding graph, p does not appear so clearly (see Haller Index / Height; p = 0.074)

Page 9:

992 subjects with repeat imaging; any therapy between imaging?

DISCUSSION

Page 10:

Perhaps this explains why most …; what kind of evaluation? Is a CT always necessary?

Page 11:

Ten-fold fewer DHS1 subjects …; Isn't it better to adjust the threshold for both indices?

In general:

Is the data of clinical relevance? Is there a predictive value from the parameters for the development of the funnel chest deformity? Should every patient undergo a CT?

Spelling:

You mixed up Haller index/Index or Correction index/Index – please unify!

6. PLOS authors have the option to publish the peer review history of their article (what does this mean?). If published, this will include your full peer review and any attached files.

Reviewer #1: Yes: Chai, Jyh-Wen

Reviewer #2: No

Reviewer #3: No

---

## [Author Response · Author response to Decision Letter 0]

14 Feb 2020

The authors wish to thank the reviewers for their thorough evaluation of the submitted manuscript. We are encouraged that the reviewers found the paper to be “one of the first of its kind.” We have addressed each of the reviewers’ comments point-by-point below.

1. Reviewer #1: As shown in Table 1, while evaluating the diagnostic indices of PE, was it necessary to include cases with pectus carinatum (PC)? The likelihood of decreased Haller index in PC patients might cause an underestimation in terms of the cases that fits the criteria.

We agree with the reviewer; the inclusion of the decreased index in pectus carinatum (PC) patients led to an underestimation of pectus excavatum (PE) cases. However, since there is a spectrum of phenotypes, including PE, PC, and mixed PE/PC, seen amongst patients referred to academic centers for pectus, we wanted to include a “real-world” patient cohort to compare against the population-based cohort. If we included only the patients who were categorized as PE by the pediatric surgeons, then the number of cases as defined by a HI>3.25, a CI>10%, or a HI>3.25 and CI>10% was 79%, 96%, and 78%, respectively. We show this graphically in Figure 3A. We have added a sentence to the Results that indicates this point (last sentence in the first Results paragraph): “A Correction index of >10% was found at the point of maximal depression in 96% of the PE cases, whereas a Haller index of >3.25 was found in fewer (78%) cases (Figure 3A).” 

2. What was the diagnostic criteria for the reference pectus cases? Apparently, not all referred cases fit both the HI and CI criteria. Providing the detailed diagnostic criteria of these cases could increase the credibility of this sample.

We have included a sentence to the Methods to clarify the criteria for the reference pectus cases. “Cases included those diagnosed with a chest wall defect by a pediatric surgeon. Only those cases with an available thoracic computed tomography (CT) scan were included in the study...” Please also refer to the response to comment #10 below.

3. While evaluating the paired samples in DHS1 and DHS2, with such a large sample size, parametric tests are usually preferred due to their higher power. Did the samples not follow normal distribution? If so, the possibility of the sample being biased may need to be addressed.

The reviewer is correct that the measurements of HI and CI, and their differences between DHS1 and DHS2, did not follow a normal distribution. That is why we chose a conservative approach to use a nonparametric test. Although nonparametric tests can be slightly less powerful than parametric tests when the distribution is indeed normal (asymptotic relative efficiency ~95.5%), they are in fact more powerful when the distribution is non-normal (Lehman EL, Nonparametrics: Statistical Methods Based on Ranks). Furthermore, none of the conclusions changed when we used a paired t-test instead (see figure below). We prefer to use the nonparametric test, because it is a more appropriate approach given the distribution of the data.

At the same time, we would like to note that the distribution of HI and CI among participants with paired data was representative of the larger DHS cohort. That is, the deviation from normality was due to the presence of a small number of extreme observations at the upper end of the distribution. Thus, the deviation from normality is unlikely to reflect some bias in the data.

4. The estimated prevalence of undiagnosed, untreated adult pectus excavatum (1 in 250, or 0.4%) was higher than the assumed incidence of neonatal pectus excavatum, 1 in 400 (0.25%), in 1975, where CT was much less available. Do the authors have any theory in regard to this phenomenon?

As mentioned below in the response to comment #7 below, we have included additional references that demonstrate the wide range of the prevalence of pectus across different cohorts. The highest rate of pectus was found in the Coskun et al study of ~1300 Turkish students between 7-14 years of age, where the prevalence of pectus carinatum was 0.6% and the prevalence of pectus excavatum was 2.6% (or, approximately 1:40). The prevalence of pectus was 1.95% in a study that evaluated ~1300 11-14 year-old students from Brazil. The Rajabi-Mashhadi study found a prevalence of chest deformities of 1% in Iranian children aged 7-14 years. Pectus is one of the most common congenital abnormalities (prevalence of 0.8%) in ~20,000 Turkish schoolchildren, 6-15 years of age. The prevalence of pectus found in this study (1:250, or 0.4%) falls within the range of prevalence described across the five studies (0.25-2.6%) (references 1-5). 

We have added text to the 2nd paragraph in the Discussion addressing this point. Cross-sectional studies have found that the Haller index tends to increase with age for pediatric populations. “Perhaps this explains why an estimates of pectus excavatum is lower in birth cohorts (1:400)[1] than in school-aged children[2,4,5] (up to 1:40[3]). Also, perhaps this is why most individuals are referred for surgical evaluation of chest wall abnormalities during adolescence.”

5. What is the clinical significance of this study? Do the authors have any expectation as to what changes this study might lead to in the management of adult PE?

We have modified the first paragraph in the Discussion to state the following: “It is striking that the radiographic measurements associated with pectus excavatum differ significantly by gender. Both the Haller and Correction indices are larger in females than males for both the case and control cohorts. Although the absolute numbers are small, we find that more females (0.5%) than males (0.3%) in the population cohort have pectus excavatum as defined by these indices. This finding is surprising since the literature suggests that a greater number of boys, usually 4-fold higher than girls, are referred for evaluation of pectus[6,19]. Even in the current study, there are 3.5 times more males than females in the pectus case cohort. Thus, there is strong referral bias to evaluate adolescent boys for pectus as breast development and modesty may mask the underlying chest wall defect in affected females. Future studies will need to determine the generalizability of these findings in unselected cohorts of different ages.” We have added the following statements to the Abstract: “Radiographic measures of pectus are more common in female than males… Despite the higher reported prevalence of pectus in adolescent males, this study demonstrates a higher prevalence of radiographic indices of pectus in adult females.”

It is unclear how these findings might lead to changes in the management of adult pectus. However, future studies could determine if there is an association between these quantitative measures with surgical or non-surgical treatment outcomes across the age spectrum.

Reviewer #2: Overall, the most important problem of this manuscript is its novelty. Similar studies have been done by other investigators before, with the same modality or other modality both. Just to list a few: Incidence and Classification of Chest Wall Deformities in Breast Augmentation Patients. Aesthetic Plast Surg (United States), Dec 2017, 41(6) p1280-1290; Relationship between cardiac MR compression classification and CT chest wall indexes in patients with pectus excavatum. J Pediatr Surg (United States), Nov 2018, 53(11) p2294-2298; Szafer D, Taylor JS, Pei A, et al.A Simplified Method for Three-Dimensional Optical Imaging and Measurement of Patients with Chest Wall Deformities. J Laparoendosc Adv Surg Tech A (United States), Feb 2019, 29(2) p267-271.

The authors respectively disagree with the reviewer on this point. The cohorts described in the references provided by the reviewer above include patients who underwent breast augmentation and those referred for evaluation of a chest wall abnormality. While multiple studies have evaluated the degree of deformity in pectus cohorts, we are not aware of one that has screened an unselected, longitudinal, multiethnic, population-based probability sampling of adults for chest wall abnormalities.

7. Reviewer #3: INTRODUCTION Page 3: First reference [1] is from 1975! Is there no newer data available?

The reference (Chung and Myrianthopoulos 1975) is the most cited reference estimating the prevalence of pectus in a birth cohort. We have added four additional references of studies that calculated the prevalence of pectus across different worldwide cohorts of children of various ages (References 2-5). The prevalence of pectus excavatum across all these studies ranges from 1:40 to 1:400. Please see our response to comment #4 above.

8. Page 3: You describe that radiographic measures can quantify …; but is it always necessary for diagnosis? What added value does CT (radiation exposure) have, especially for patients in adolescence. A small paragraph like diagnosis is based on … would be helpful.

We agree with the reviewer regarding this point. We have added a sentence in the Discussion (4th paragraph, 1st sentence) that “Although a CT scan of the chest is not needed to make the diagnosis of pectus, measurements from the imaging study offer metrics for comparison of cases against controls.”

9. Page 3: A Correction index of 10% - do you mean exactly or greater than? 

We have edited the sentence as follows “a Correction index of greater than 10% is considered indicative of substantial pectus excavatum.”

10. METHODS Page 4: The cases referred for evaluation of pectus …; During which period were the patients selected? How many cases were obtained and how many were included in the study? Why did all patients undergo a CT? 

Patients were recruited by East Virginia Medical Center from 2009-2017; and by UTSW from 2014-2017. Only patients with a CT scan of the chest that was available for review were included as cases in this manuscript. Thus, fewer patients were included as clinical cases in this study (58% and 17% of the total cohort collected from Virginia and UTSW, respectively). We have added these details to the Methods section.

While we assume that the chest CT scan was ordered to further evaluate the chest wall deformity, we were not able to determine the clinical indication(s) for the chest CT scans for all patients recruited from both centers. 

11. Page 5: Definition of Haller and Correction index thresholds - taken from the literature? 

Yes. The references are now included in the Methods section describing the index thresholds.

12. RESULTS Page 7: The Haller index measures … The Correction index measures …; just a repetition of the introduction / methods - can be shortened. 

Thank you for the comment. We have shortened this section.

13. Page 8: You describe significant associations; do the parameters mentioned have a predictive value? 

To determine if the Haller and Correction indices have a predictive value in distinguishing pectus excavatum cases from the general population, we have performed ROC curve analyses (figures shown below). For this analysis, we excluded the pectus carinatum and mixed cases, and compared clinical pectus excavatum cases to the DHS (controls). Pectus excavatum cases with HI > 3.25 and CI > 0.1 were considered as true positives. DHS participants with indices below these thresholds were considered as true negatives. Overall, HI and CI parameters were able to discriminate between PE cases and DHS controls quite well, with areas under the ROC curve (AUROC) greater than 97% for HI and CI measured at the level of superior xiphoid (SX). CI >10% at the level of SX had the best performance, with sensitivity and specificity both greater than 94%. Although these results look promising, we would like to point out that we consider them preliminary, since the comparison populations were very different with respect to age, gender, and ethnicity. Future studies will be required to validate these findings before the HI and CI parameters can be used in the clinic for routine screening or diagnosis of subjects.

14. Page 9: correlated with age (p=0.0045); for real? Or 0.045?; if you look at the corresponding graph, p does not appear so clearly (see Haller index / Height; p = 0.074) 

Yes, the p-value as indicated (p=0.0045) is correct. The trend is very modest, but given the sample size, we can detect even a small non-zero trend with relatively high level of significance.

15. Page 9: 992 subjects with repeat imaging; any therapy between imaging? 

There was no chest wall specific therapy between imaging. In fact, we excluded DHS scans if there was evidence of chest trauma or a prior thoracic surgical procedure (described in Methods).

16. DISCUSSION Page 10: Perhaps this explains why most …; what kind of evaluation? Is a CT always necessary? 

We have revised the sentence to improve its clarity. Comment #4 (above) also described the modification of this sentence. 

17. Page 11: Ten-fold fewer DHS1 subjects …; Isn't it better to adjust the threshold for both indices? 

We agree with the reviewer that we could adjust the threshold in adults for both indices based upon their distribution found in the normal cohort. A threshold of greater than 2 standard deviations from the mean would be equivalent to a HI > 2.759 and a CI > 0.117. (Under the assumption of a normal distribution, approximately 2.5% of samples would be expected to exceed this value. In DHS-1, 3.46% and 3.42% exceed the threshold for HI and CI, respectively, reflecting the presence of extreme observations and a deviation from a normal distribution.) A threshold of greater than 3 standard deviations from the mean would be equivalent to a HI > 3.05 and a CI > 0.152. (In DHS-1, 0.93% and 1.07% exceed these thresholds, compared to 0.135% expected under the normal distribution). While these thresholds would be applicable to adult multiethnic cohorts, they would not be applicable to adolescent pectus case cohorts. 

18. In general: Is the data of clinical relevance? Is there a predictive value from the parameters for the development of the funnel chest deformity? Should every patient undergo a CT? 

This question is very similar to question #5 posed by Reviewer #1. Kindly refer to comment #5.

19. Spelling: You mixed up Haller index/Index or Correction index/Index – please unify!

We have revised the text so that capitalization of the word “index” is uniform.

Thank you.

Best regards,

Christine Kim Garcia, MD, PhD

---

## [Decision Letter · Decision Letter 1]

20 Apr 2020

Prevalence of Pectus Excavatum in an Adult Population-Based Cohort Estimated from Radiographic Indices of Chest Wall Shape

PONE-D-19-18160R1

Dear Dr. Kim Garcia,

We are pleased to inform you that your manuscript has been judged scientifically suitable for publication and will be formally accepted for publication once it complies with all outstanding technical requirements.

With kind regards,

JJ Cray Jr., Ph.D.

Academic Editor

PLOS ONE

Additional Editor Comments (optional):

Reviewers' comments:

Reviewer's Responses to Questions

**Comments to the Author**

1. If the authors have adequately addressed your comments raised in a previous round of review and you feel that this manuscript is now acceptable for publication, you may indicate that here to bypass the “Comments to the Author” section, enter your conflict of interest statement in the “Confidential to Editor” section, and submit your "Accept" recommendation.

Reviewer #1: All comments have been addressed

Reviewer #2: All comments have been addressed

Reviewer #3: All comments have been addressed

2. Is the manuscript technically sound, and do the data support the conclusions?

Reviewer #1: Yes

Reviewer #2: Yes

Reviewer #3: Yes

3. Has the statistical analysis been performed appropriately and rigorously? 

Reviewer #1: Yes

Reviewer #2: N/A

Reviewer #3: Yes

4. Have the authors made all data underlying the findings in their manuscript fully available?

Reviewer #1: Yes

Reviewer #2: Yes

Reviewer #3: Yes

5. Is the manuscript presented in an intelligible fashion and written in standard English?

Reviewer #1: Yes

Reviewer #2: Yes

Reviewer #3: Yes

6. Review Comments to the Author

Reviewer #1: (No Response)

Reviewer #2: (No Response)

Reviewer #3: (No Response)

7. PLOS authors have the option to publish the peer review history of their article (what does this mean?). If published, this will include your full peer review and any attached files.

Reviewer #1: Yes: Chai, Jyh-Wen

Reviewer #2: No

Reviewer #3: No

---

## [Editor Report · Acceptance letter]

28 Apr 2020

PONE-D-19-18160R1 

Prevalence of Pectus Excavatum in an Adult Population-Based Cohort Estimated from Radiographic Indices of Chest Wall Shape 

Dear Dr. Kim Garcia:

I am pleased to inform you that your manuscript has been deemed suitable for publication in PLOS ONE. Congratulations! Your manuscript is now with our production department. 

With kind regards,

on behalf of

Dr. JJ Cray Jr. 

Academic Editor

PLOS ONE